# The COVID-19 Pandemic Is Associated with Reduced Survival after Pancreatic Ductal Adenocarcinoma Diagnosis: A Single-Centre Retrospective Analysis

**DOI:** 10.3390/jcm11092574

**Published:** 2022-05-04

**Authors:** Oliver Madge, Alexandra Brodey, Jordan Bowen, George Nicholson, Shivan Sivakumar, Matthew J. Bottomley

**Affiliations:** 1Ambulatory Assessment Unit, Oxford University Hospitals NHS Foundation Trust, Oxford OX3 9DU, UK; oghm2-mdpi@srcf.net (O.M.); jordan.bowen@ouh.nhs.uk (J.B.); 2Department of Oncology, Oxford University Hospitals NHS Foundation Trust, Oxford OX3 9DU, UK; alex.brodey@ouh.nhs.uk (A.B.); shivan.sivakumar@oncology.ox.ac.uk (S.S.); 3Department of Statistics, University of Oxford, Oxford OX1 3LB, UK; george.nicholson@stats.ox.ac.uk; 4Department of Oncology, University of Oxford, Oxford OX3 7DQ, UK; 5CAMS Oxford Institute, Nuffield Department of Medicine, University of Oxford, Oxford OX3 7FZ, UK

**Keywords:** pancreatic cancer, COVID-19, outcomes, treatment, survival

## Abstract

The COVID-19 pandemic has hugely disrupted healthcare provision, including oncology services. To evaluate the effects of the pandemic on referral routes leading to diagnosis, treatments, and prognosis in patients with pancreatic ductal adenocarcinoma, we performed a retrospective cohort study at a single tertiary centre in the UK. The patients were identified from the weekly hepatopancreatobiliary multidisciplinary team meetings between February 2018 and March 2021. The demographic, referral, and treatment data for each patient and date of death, where applicable, were extracted from the electronic patient record. The patients (*n* = 203) were divided into “pre-pandemic” and “pandemic” cohorts based on a referral date cut-off of 23rd March 2020. The median survival was 7.4 months [4.9–9.3] in the “pre-pandemic” cohort (*n* = 125), halving to 3.3 months [2.2–6.0], (*p* = 0.015) in the “pandemic” cohort (*n* = 78). There was no significant difference in patient characteristics between the two cohorts. There was a trend toward increased emergency presentations at diagnosis and reduced use of surgical resection in the “pandemic” cohort. This small-scale study suggested that the COVID-19 pandemic is associated with a halving of median survival in pancreatic ductal adenocarcinoma. Urgent further studies are required to confirm these findings and examine corresponding effects in other cancer types.

## 1. Introduction

Pancreatic ductal adenocarcinoma (PDAC) has the worst prognosis of any of the common cancers, with a 1-year survival rate of 25.9% and a 5-year survival rate of 7.3% in the UK [1]. In common with other cancers, survival is improved with an earlier diagnosis of localised disease [2,3].

The emergence and rapid worldwide dissemination of COVID-19 from late 2019 onwards has caused major disruption to healthcare systems globally [4], with a significant impact on health services for noncommunicable diseases [5]. In April 2020, two-week-wait (2WW) referrals from primary care for upper-gastrointestinal malignancy fell nationally in England by 65% compared to April 2019 [6], and modelling studies predicted a significant effect of reduced referrals on cancer survival [7].

There are various routes through the healthcare system that patients follow to arrive at a diagnosis of PDAC, broadly categorised as nonemergency or emergency presentations. Nonemergency pathways include direct referral from primary or secondary care due to symptomatology or clinician concern or referral for an incidental finding during investigation of an unrelated complaint. Emergency pathways include self-presentation to emergency departments, primary care services, or secondary care services (for example same-day emergency care units).

Data from the UK National Cancer Intelligence Network (NCIN) Routes to Diagnosis project show that PDAC is diagnosed following emergency presentation twice as often as other cancers (47%) [8]. Referrals leading to a diagnosis of PDAC for the period 2006–2015 [8] were 36% from nonemergency primary care (two-week-wait (2WW) or other referrals) and 47% from emergencies, with the remainder coming from other outpatient, inpatient elective, postmortem diagnosis, or unknown routes [8]. Early, nonemergency PDAC diagnosis via the 2WW pathway is associated with three times the one-year survival compared to emergency presentation [8,9].

PDAC presents a particularly interesting target for research into the changes in routes to diagnosis and how the unprecedented circumstances of the COVID-19 pandemic affect prognosis. Due to the indolence of its presentation, it is a malignancy at high risk of being overlooked among the healthcare challenges of a pandemic. Given its poor prognosis, it would be expected that any effect on mortality would become apparent relatively quickly. We therefore undertook an observational study to assess the impact of the pandemic upon PDAC outcomes. We hypothesised that, during the COVID-19 pandemic, there would be an increased proportion of emergency presentations leading to a diagnosis of PDAC, that treatment availability would be restricted, and that survival from referral would, therefore, be reduced.

## 2. Materials and Methods

This study was reported in accordance with the “Strengthening Reporting of Observational Studies in Epidemiology” (STROBE) guidelines (completed checklist in Appendix A) [10].

## 3. Study Design and Setting

The study utilised a retrospective cohort design. The setting was the Oxford University Hospitals NHS Foundation Trust (OUHFT), a tertiary oncology centre in the southeast of England. OUHFT coordinates the hepatopancreatobiliary (HPB) multidisciplinary team (MDT) for a regional network of other Trusts. Referrals for investigation and management of PDAC come from within the Trust secondary care catchment area, as well as the wider region. As a retrospective audit, this study was exempt from requiring NHS research ethics approval or informed consent of participants.

## 4. Participants

The inclusion criteria for the study were a diagnosis of PDAC and a home postcode in the secondary care catchment area of OUHFT. Patients not fulfilling these criteria were excluded.

Potential study participants were drawn from a list of patients discussed in the Oxford (local) section of a weekly meeting of the OUHFT HPB MDT between 1st February 2018 and 31st March 2021, inclusive. The medical specialists attending these meetings include oncologists, gastroenterologists, hepatopancreatobiliary surgeons, radiologists, and histopathologists, alongside clinical nurse specialists. The diagnosis of PDAC was either confirmed with histology or on the expert opinion of an oncologist specialising in PDAC (S.S.). In order to minimise ascertainment bias due to loss of follow-up, patients with a home postcode outside the secondary care catchment area of OUHFT were excluded.

## 5. Data Collection

Patient identifiers were programmatically extracted from the MDT meeting discussion lists and then used to retrieve the coded diagnosis, demographics, and referral source of each patient from the hospital electronic patient record system. Duplicates and those not meeting the inclusion criteria were excluded. The clinical records for each included patient were manually examined by a member of the study team (O.M. or A.B.) to confirm the PDAC diagnosis.

Follow-up was via notes review on the Trust electronic patient record system. Follow-up was until death or 28 December 2021 (whichever was earlier). The hospital chemotherapy prescribing system was manually examined by the research team (A.B. and S.S.) to determine each patient’s stage and performance status at diagnosis.

To calculate the stage of the disease, imaging reports, MDT reports, and clinic letters were reviewed. The patients were classified as Stage 1–4 based on the AJCC eighth edition staging of pancreatic cancer [11].

The performance status at diagnosis was preferentially extracted from the MDT notes. Where the MDT notes did not refer to performance status, the primary care referral to the MDT was used as the data source; if the referral information did not include the patient’s performance status, then it was taken from the earliest clinic letter following diagnosis. Where performance status was not specifically stated in any of the preceding sources, it was calculated from any available documentation in the electronic record system, including inpatient notes, clinic letters, and MDT notes.

All the included patients were reviewed with regards to whether they were offered chemotherapy as part of their management. Data on whether patients were offered chemotherapy, which regimens they were offered, and if they received chemotherapy were recorded. Where chemotherapy was both offered and given, the regimens given and start dates were collected from the hospital’s chemotherapy prescribing system.

The clinical data collected included age at referral, sex, date of referral, home postcode, diagnosis, ethnicity, stage at diagnosis, performance status at diagnosis, date of death, referral source, whether a surgical resection was offered, whether surgical resection was performed, whether an oncology appointment was offered, whether chemotherapy was offered, and the type of chemotherapy received.

To exclude COVID-19-related mortality as a reason for poorer outcomes in the “pandemic” cohort, we assessed COVID-19 status at death. Where the death certificate was issued by the hospital, this was examined for mention of COVID-19 as a primary or secondary cause of death. If the death certificate was not available, a SARS-CoV-2 PCR swab result from within 7 days of death was used to estimate the COVID-19 status at death [12]. Where neither were available, the presence of a clinical review on the electronic patient record recorded within 7 days prior to death was used as an indicator of likely COVID-19 status at death. Seven days was chosen to be the cut-off for a SARS-CoV-2 PCR swab or clinical review, as COVID-19 contracted after that time is unlikely to be the cause of death [13]. If none of the preceding were available, the COVID-19 status at death was recorded as indeterminate.

As an exploratory and retrospective study, all eligible cases during the analysis period were included, and no prespecified sample size calculation was performed.

## 6. Analysis

The study patients were divided into two cohorts by referral date. The nominal start of the COVID-19 pandemic in England was taken as the announcement of the first England-wide lockdown on the 23rd of March 2020. Patients referred prior to this date were classified as “pre-pandemic”, and patients referred on or after this date were classified as “pandemic”.

Demographic indicators, tumour stage, patient performance status at diagnosis, referral source, whether surgery was offered or received, whether the patient was seen in the oncology clinic, whether chemotherapy was offered, and the type of chemotherapy received between the two groups were analysed for statistically significant differences. Sex, ethnicity, disease stage, performance status, referral source, surgery, and chemotherapy type (categorical data) were compared between the “pre-pandemic” and “pandemic” cohorts using chi-squared testing; age at referral was similarly compared using a two-tailed Student’s *t*-test. The categorical data were reported as numbers (percentage), whilst continuous data were reported as medians (interquartile range).

The mortality data were analysed using standard methods of Kaplan–Meier survival analysis to calculate the median survival and the restricted mean survival time (RMST) to estimate the change in survival time over one year [14]. For the RMST analysis, tau was set to one year, as this encompassed the follow-up for the majority of patients, thereby minimising censoring in the analysis.

To explore further the associations between the patient variables and survival, the data were used to construct a Cox proportional hazards regression model. The explanatory variables were age at referral (in years), sex, disease stage at diagnosis, performance status, referral source, whether surgery was received, whether chemotherapy was offered, and whether the referral was prior to or during the pandemic. Univariate analyses were performed for each variable. A multivariate analysis was performed with all the explanatory variables included in the model.

## 7. Results

### 7.1. Demographics and Disease Status at Diagnosis

There were 1873 patient identifiers extracted from the HPB MDT lists during the study period. Of these, 1164 were not cases of PDAC, and 417 were duplicates, leaving 292 patients whose electronic patient records were examined. Of these, a further 61 cases were found not to be PDAC, and 39 had an address outside the catchment area (11 were excluded on both criteria). This left 203 eligible patients to be included in the analysis (see also the flowchart in Appendix A). A total of 125 (62%) were referred prior to the nominal start of the COVID-19 pandemic and fell within the “pre-pandemic” cohort, while 78 (38%) fell within the “pandemic” cohort.

The demographic and referral information is shown in Table 1. The major demographic characteristics that might impact disease survival did not differ between the two cohorts. Similarly, there were no significant differences between the disease stage or performance status at diagnosis.

### 7.2. Origin of Diagnostic Referral

There was a nonsignificant reduction in the proportion of referrals originating from primary care during the “pandemic” compared to the “pre-pandemic” period (*p* = 0.14) (Figure 1). In the pre-pandemic period, there were 4.9 ± 2.5 referrals per month. This increased slightly to 6.3 ± 2.2 per month during the pandemic.

### 7.3. Patient Survival from Diagnosis

A total of 177 patients (87%) died during follow-up: 111 patients (89%) from the “pre-pandemic” cohort and 66 (85%) from the “pandemic” cohort. The median follow-up time for the whole cohort was 5.5 months [1.7–12.9] (to death or censoring). Strikingly, there was a halving of the median survival in the “pandemic” cohort compared to the “pre-pandemic” cohort (7.4 months [4.9–9.3] vs. 3.3 months [2.2–5.9], *p* = 0.015) (Figure 2A). The restricted mean survival time analysis suggested a 1.8-month [0.5–3.0] reduction in survival at one year for the pandemic cohort (*p* = 0.006).

The disease stage at diagnosis is highly predictive of subsequent survival in PDAC [2]. When we compared the median survival for each disease stage at diagnosis for the two cohorts, we found a reduction in the median survival in disease stages 2, 3, and 4 in the “pandemic” cohort, with the largest reductions in stages 2 and 3 (Figure 2B). Stage 1 disease showed a 3-month increase in the median survival period in the “pandemic” cohort. Reductions in the median survival by 9.8 months, 3.8 months, and 1 month in the “pandemic” cohort were seen in stages 2, 3, and 4, respectively.

COVID-19 was not a major contributor to mortality based on death certification (*n* = 31), SARS-CoV-2 PCR (*n* = 6), or COVID-19 reference by palliative care review (*n* = 23) within seven days of death (Table 2). Overall, three patients in the “pandemic” cohort were known to have COVID-19 at the time of death (4.9% of deaths where information was available). Excluding these patients and rerunning the survival analysis did not alter the median survival or RMST conclusions.

### 7.4. Treatment Offered

Surgical resection was offered only in a minority of cases in both cohorts. A total of 27 patients (22%) of the “pre-pandemic” cohort were offered surgery compared to 8 (10%) of the “pandemic” cohort (*p* = 0.059). This is broken down by disease stage at diagnosis in Figure 3A. Of the patients with stage 1 or 2 disease, 79% of the “pre-pandemic” cohort was offered surgery compared to 45% of the “pandemic” cohort (*p* = 0.046).

A total of 15 (12%) patients in the “pre-pandemic” cohort went on to receive surgery compared to 4 (5%) patients in the “pandemic” cohort (*p* = 0.165) (Figure 3B shows this by stage). In patients with stage 1 or 2 disease, 46% of the “pre-pandemic” cohort went on to receive surgery compared to 27% of the “pandemic” cohort (*p* = 0.504).

Oncology clinic appointments were arranged for 93 (74%) of the “pre-pandemic” cohort compared to 48 (62%) of the “pandemic” cohort (*p* = 0.075) (Figure 3C shows this by stage). Where an oncology appointment was not arranged, this was due either to a recommendation for palliative care from the MDT or patient choice. Chemotherapy was offered to 72 (58%) of the “pre-pandemic” cohort and 37 (47%) of the “pandemic” cohort (*p* = 0.205) (Figure 3D shows this by stage). A total of 42 (34%) of the “pre-pandemic” cohort received FOLFIRINOX or modified FOLFIRINOX compared to 28 (36%) patients in the “pandemic” cohort (*p* = 0.855). A total of 23 (18%) of the “pre-pandemic” cohort received gemcitabine (single agent or in combination) compared to 13 (17%) patients in the “pandemic” cohort (*p* = 0.900).

### 7.5. Cox Regression Model

The hazard ratios and *p* values from the univariate and multivariate models are shown in Table 3. The univariate model confirmed an association between a referral being made during the pandemic and an increased hazard ratio of 1.47 [1.07–2.01] (*p* = 0.016). However, when including all the other explanatory variables in the multivariate model, referral during the pandemic was no longer a significant independent predictor of survival, with a hazard ratio of 1.29 [0.94–1.77] (*p* = 0.110).

## 8. Discussion

The COVID-19 pandemic has put extraordinary strain on healthcare systems, leading to changes in public health messaging, patient behaviour, and healthcare resources and availability. We demonstrated here that PDAC survival from diagnosis worsened significantly during the pandemic compared to the pre-pandemic period, with a halving of median survival. All areas of healthcare have been affected by the pandemic and, therefore, the possibility of altered outcomes should be considered across the spectrum of malignant and nonmalignant conditions.

We did not demonstrate any single factor that could adequately explain the survival difference in isolation; rather, we postulated that the causes were multifactorial, including increased likelihood of emergency presentation, reduced use of surgical resection, changes in treatment regimes, and other possible factors that we were not able to explore in this study. This hypothesis was supported by a Cox regression model that demonstrated a reduced significance in referral during the pandemic itself as an independent predictor of survival once the other explanatory variables analysed in the study were included in a multivariate model.

Notably, the disease stage and performance status at diagnosis did not differ between the two cohorts in this study. It is possible that poorer survival was fully explained by the efforts to reduce risk to patients from COVID-19. In previous data from our centre, one-third of preferred hepatopancreatobiliary MDT recommendations for surgery were altered due to concerns about the risk to the patient in proceeding with treatment in the context of the pandemic or due to the presumed balance of risk and benefit in patients with borderline performance status [15]. Surgical resection offers the only possibility of a cure in PDAC [16], and any restriction of surgical treatment is, therefore, likely to lead to a reduction in overall survival. We demonstrated a trend towards reduced use of surgery in the “pandemic” cohort, although the proportions receiving surgery in both cohorts were small.

An alternative explanation for our findings is that they reflected a delayed presentation of patients with undiagnosed PDAC during the pandemic, leading to lead-time bias, where the granularity of the staging system was unable to capture the subtleties of patient disease and suitability for treatment. This is especially pertinent to PDAC, where rapid disease progression during any delay in presentation can affect a patient’s suitability for treatment [17]. The potential for delay in PDAC diagnosis through the pandemic has been multi-faceted, including late self-presentation of patients experiencing vague, nonspecific symptoms typical of PDAC whose overriding concern was not overwhelming a fragile healthcare system [18]; availability of appointments in primary care [19]; and lack of capacity in secondary care [20].

One further possible explanation for reduced survival in the “pandemic” cohort could be significant mortality directly due to COVID-19 infection, as the survival analysis was based on all-cause mortality. We were able to infer the COVID-19 status at death for 92% of the “pandemic” cohort, and despite inherent limitations with inference, we did not identify a significant proportion of deaths from COVID-19 in the “pandemic” cohort with survival unaffected by the exclusion of the confirmed cases. The UK government strongly promoted a policy of “shielding” for clinically vulnerable patients from March 2020 until April 2021, including those with cancer, whereby they were advised to minimize contact with others to reduce the risk of contracting COVID-19. There was very strong adherence to this advice [21] and, therefore, in the context of the data we were able to collect on COVID-19 status at death, the likelihood of significant numbers of patients in the “pandemic” cohort having died with COVID-19 was low.

A consensus position paper based on the professional opinions of UK clinicians treating PDAC was published in July 2020 [22]. This recommended prioritising treatments that limited hospital visits and promoted the use of hypofractionated precision radiotherapy and altered chemotherapy regimes, in some cases as a bridge to surgery of potentially resectable disease. Although this approach was taken in a pragmatic effort to balance the overall risk to patients, it has the potential to affect survival by substituting the best-known treatment for one with inferior outcomes.

To date, there have been limited publications directly investigating associations between the COVID-19 pandemic and survival in patients with malignancy. Our findings are in keeping with a modelling study published near the beginning of the pandemic that looked at breast, colorectal, oesophageal, and lung cancers and predicted an increase in deaths at 5 years [23]. Lai et al. estimated excess mortality across a range of malignancies under various scenarios [24]. They modelled the excess indirect mortality of patients with malignancy due to service disruption as a hazard ratio and applied this to a proportion of the population assumed to be affected. Whilst their modelling assumed a 10% COVID-19 infection rate, the model allowed patients to be assigned low risk from COVID-19 mortality (modelled as having no comorbidities in addition to malignancy), which removed the effect of this unrealistic assumption. Using their online modelling tool to look specifically at PDAC [25], a hazard ratio of 1.2 and 80% affected population would produce the 1-year mortality of 75% seen in this study; a hazard ratio of 1.4 would produce the same mortality estimate under the assumption of 40% affected population.

A Portuguese study retrospectively examined the effect of the pandemic on the short-term survival of patients with multiple types of cancer treated at the Portuguese Oncology Institute of Porto in the early pandemic (March 2020 to July 2020) with a follow-up to 4 months [26]. This demonstrated an increased risk of death in patients with stage 3 disease during the pandemic compared to pre-pandemic, as well as those undergoing surgery or receiving radiotherapy.

With the exception of the Portuguese study, we are not aware of any other published studies that examine the association between the pandemic and mortality in cancer using real-world data.

The strengths of this study include complete follow-up for the whole cohort. There were no significant differences in sex, age, or ethnic background between the two study cohorts, suggesting the underlying populations were similar, though confounders cannot be completed excluded. This was not unexpected given the relatively short period of the study, the consecutive time periods of the two groups, and the localisation to a confined geographical area.

Whilst this study provided novel and important data regarding the impact of COVID-19 upon patient outcomes in PDAC, there were some limitations. This was a single-centre study carried out in a well-defined geographical area with access to healthcare facilities that may or may not be replicated elsewhere. The sample size of the study was limited. Notably, the catchment area of OUHFT demonstrated overrepresentation of Caucasian ethnic groups and a relatively low rate of socio-economic deprivation [27]. Whilst ethnicity is not known to affect outcomes in PDAC, socio-economic deprivation is associated with poorer prognosis [28]. A potential bias could arise from missed cases of PDAC in the study population. The use of MDT data to identify cases provided a consistent point of entry to diagnosis of PDAC. However, the subsequent analysis relied on coding data, and errors here could lead to a loss of cases. This would not be expected to introduce systemic bias as it is likely to be random, but it may have been more pronounced during the pandemic period when staff pressures were increased.

The follow-up time for the “pandemic” cohort was short (March 2020 to December 2021 in the worst case), though this led to only two censored events at one-year survival. We would not, therefore, expect to see a significant effect on the median survival or the RMST analysis with a longer follow-up.

In conclusion, our findings suggested the COVID-19 pandemic is associated with a halving of the median survival from diagnosis in PDAC compared to the pre-pandemic period. Whilst it is not unreasonable to assume that other cancers may show a similar effect, it is likely to be particularly notable in PDAC (and, therefore, detectable at an earlier stage following the start of the pandemic) due to the overall poor survival and the marked impact of early intervention upon outcomes. The reasons for the findings were unclear, in particular the relative weights of the proposed explanatory factors on the survival difference between the study cohorts. Larger multicentre studies are urgently needed to validate and expand our results to include investigation of other forms of cancer in wider geographical settings and to understand more clearly the underlying contributory factors. It is vitally important to learn lessons from the unprecedented event of the COVID-19 pandemic in order to adequately plan for and respond to future events with the potential to cause interruption to healthcare services.

## Figures and Tables

**Figure 1 jcm-11-02574-f001:**
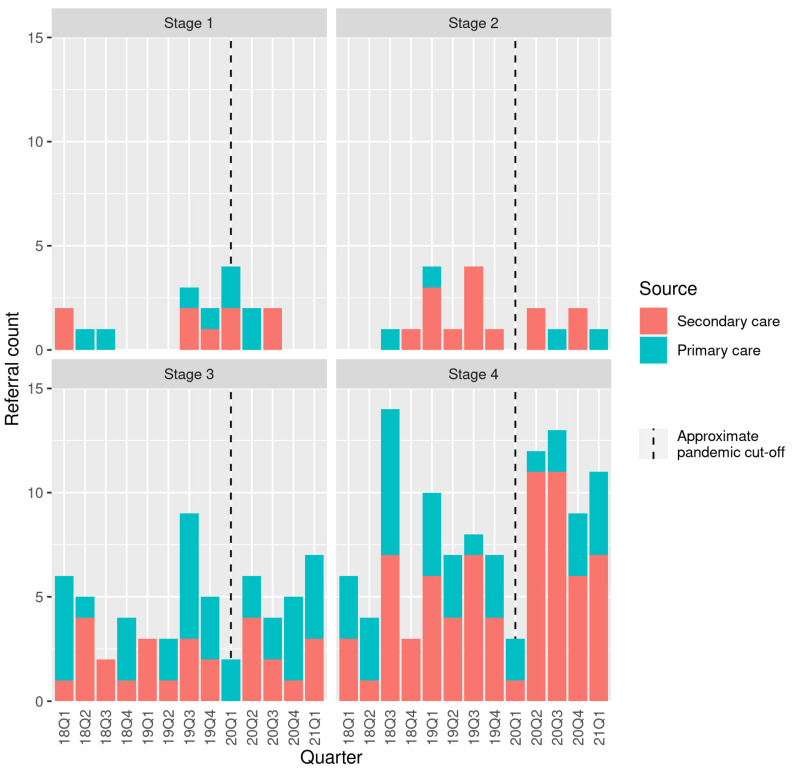
Referrals to the MDT meeting leading to a diagnosis of pancreatic ductal adenocarcinoma (PDAC) stratified by disease stage at diagnosis.

**Figure 2 jcm-11-02574-f002:**
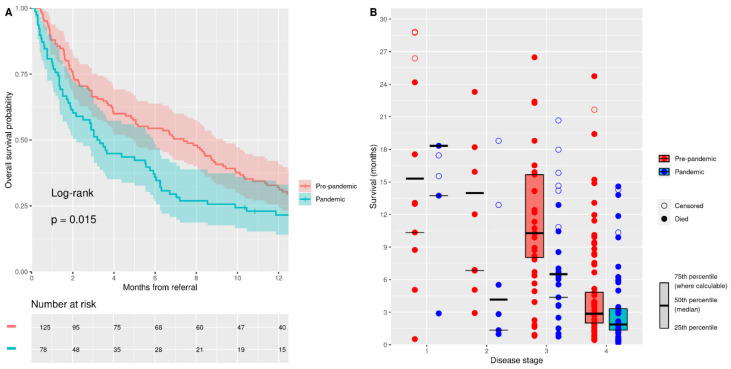
Survival analysis (all causes of mortality) from referral. (**A**) Kaplan–Meier plot of survival from referral (all disease stages at diagnosis). (**B**) Box plot of survival from referral in “pandemic” and “pre-pandemic” cohorts by disease stage at diagnosis. Individual cases plotted and IQR shown where the 75th percentile is calculable.

**Figure 3 jcm-11-02574-f003:**
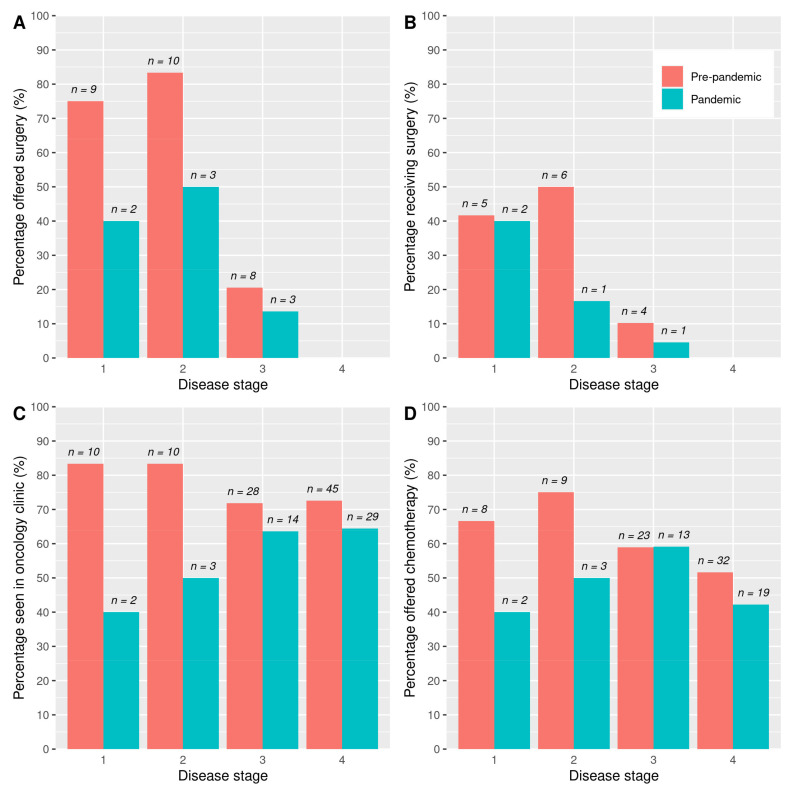
Treatments offered and received by stage at diagnosis. (**A**) Percentage (number) of patients offered surgery. (**B**) Percentage (number) of patients who received surgery. (**C**) Percentage (number) of patients seen in the oncology clinic. (**D**) Percentage (number) of patients offered chemotherapy.

**Table 1 jcm-11-02574-t001:** Characteristics of patients diagnosed with PDAC.

	Pre-Pandemic (*n* = 125)	Pandemic (*n* = 78)	
Gender			
Female	64 (51%)	39 (50%)	*p* = 0.98
Male	61 (49%)	39 (50%)	
Age			
Median	72 years	76 years	*p* = 0.28
IQR	17 years	16 years	
Ethnic background			
Any White	89 (71%)	56 (72%)	*p* = 0.25
Any Asian	2	1	
Any Black	2	0	
Any mixed or other	0	2	
Unknown or not stated	32 (26%)	19 (24%)	
Stage			
Stage 1	12 (10%)	5 (6%)	*p* = 0.68
Stage 2	12 (10%)	6 (8%)	
Stage 3	39 (31%)	22 (28%)	
Stage 4	62 (50%)	45 (58%)	
Performance status			
PS 0	42 (34%)	21 (27%)	*p* = 0.44
PS 1	40 (32%)	31 (39%)	
PS 2	30 (24%)	16 (21%)	
PS 3	11 (9%)	10 (13%)	
PS 4	2 (2%)	0	
Referral source			
Primary care	56 (45%)	26 (33%)	*p* = 0.14
Secondary care	69 (55%)	52 (67%)	

Age significance value by two-sample *t*-test. All other significance values by chi-squared test.

**Table 2 jcm-11-02574-t002:** Information sources used to infer COVID-19 status at death for the “pandemic” cohort.

	COVID-19 +ve	COVID-19 −ve
Death certificate	2	29
SARS-CoV-2 PCR ^1^	0	6
Palliative care review ^1^	1	23
Total ^2^	3 (4.9%)	58 (95.1%)

^1^ Within 7 days of death. ^2^ No information was available for 5 patients.

**Table 3 jcm-11-02574-t003:** Cox proportional hazards model for survival in patients diagnosed with PDAC.

	Univariate Model		Multivariate Model	
	Hazard Ratio	*p* Value	Hazard Ratio	*p* Value
Age at referral	1.02 (1.01–1.04)	0.002	1.01 (1.00–1.03)	0.149
Male sex	1.00 (0.74–1.34)	0.995	1.14 (0.83–1.56)	0.411
Stage	1.72 (1.42–2.07)	<0.001	1.64 (1.33–2.02)	<0.001
Performance status	1.62 (1.39–1.89)	<0.001	1.28 (1.07–1.53)	0.008
Primary care referral	0.75 (0.56–1.02)	0.067	0.76 (0.55–1.04)	0.086
Surgery received	0.15 (0.07–0.31)	<0.001	0.31 (0.14–0.68)	0.003
Chemotherapy offered	0.33 (0.25–0.45)	<0.001	0.50 (0.34–0.74)	0.001
Referral during pandemic	1.47 (1.07–2.01)	0.016	1.29 (0.94–1.77)	0.110

## Data Availability

The data presented in this study are available on request from the corresponding author. The data are not publicly available due to institutional privacy.

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
