# Peer review of "The COVID-19 Pandemic Is Associated with Reduced Survival after Pancreatic Ductal Adenocarcinoma Diagnosis: A Single-Centre Retrospective Analysis"

_jcm, 2022, doi:10.3390/jcm11092574_

Round 1

Reviewer 1 Report

The paper is well written and the research topic is of great interest and modernity.

I have the following remarks.

1) The research was carried out in a well-defined geographical area with unique health assistance facilities that may or may not be replicated elsewhere. I think this limitation should be better emphasized in the title as well as in the discussion.

2) The authors assumed that the inferior survival in the pandemic cohort may be attributable to worse access to surgery, chemotherapy, and maybe to the hospital facility as a whole (people were told to stay at home in those days). Is it possible to evaluate which of these factors weighed more than the others?

3) The ultimate cause of death has not been elucidated. Is it possible that a certain amount of deaths of these pancreatic cancer patients might have been due to COVID itself? If this had been the case, then the entire scenario should be reinterpreted.

Reviewer 2 Report

The COVID-19 pandemic is associated with reduced survival

after pancreatic ductal adenocarcinoma diagnosis: a single-centre retrospective analysis - Madge et al. 2022

COMMENTS TO THE AUTHORS:

Thank you for giving me the chance to review this manuscript which addresses a clinically relevant topic. The COVID-19 pandemic has huge impact on the worldwide field of oncology.

Overall:

  • I’m very happy with the fact that you used the STROBE guidelines

Abstract:

  • Please mention the number of included patients, total, pre and pandemic

Introduction:

  • No comments

Methods:

  • Please describe which specialties attend the Oxford (local) section of a weekly meeting of the OUHFT HPB MDT

Results:

  • Please mention the % of ‘’event’’ or death in both cohort during the study time ‘’ died during follow-u’’
  • I think page 6 is incomplete ‘’ When we compared median survival and interquartile range for each disease stage at diagnosis for the two cohorts, we found a reduction in median sur…..’’

Discussion

  • The follow up of the pandemic cohort is short (march 2020 – December 2021) please mention

I want to congratulate the authors with their hard work and nice paper.
